# CODEALIGNBENCH : ASSESSING CODE GENERATION MODELS ON DEVELOPER-PREFERRED CODE ADJUSTMENTS

## ABSTRACT

As large language models become increasingly capable of generating code, evaluating their performance remains a complex and evolving challenge. Existing benchmarks primarily focus on functional correctness, overlooking the diversity of real-world coding tasks and developer expectations. To this end, we introduce a multi-language benchmark that evaluates LLM instruction-following capabilities and is extensible to operate on any set of standalone coding problems. Our benchmark evaluates instruction following in two key settings: adherence to *pre-defined* constraints specified with the initial problem, and the ability to perform refinements based on *follow-up* instructions. For this paper's analysis, we empirically evaluated our benchmarking pipeline with programming tasks from LiveBench, that are also automatically translated from Python into Java and JavaScript. Our automated benchmark reveals that models exhibit differing levels of performance across multiple dimensions of instruction-following. Our benchmarking pipeline provides a more comprehensive evaluation of code generation models, highlighting their strengths and limitations across languages and generation goals.

## 1 INTRODUCTION

Program synthesis has been a long standing challenge in the field of computer science research. It is defined as the automatic generation of programs in a given language in order to fulfill user intent, typically expressed through natural language instructions (Gulwani et al., 2017). It is a particularly difficult challenge because user intents are often underspecified, ambiguous, or expressed in ways that leave multiple valid interpretations. This is compounded by the fact that for any given user intent, there may exist a vast search space of programs that are both syntactically correct and semantically valid, which increases the complexity of finding a solution that precisely matches the user's desired behavior.

This challenge has grown more attainable with the advent of Large Language Models (LLMs). LLMs have demonstrated impressive capabilities in various code generation tasks, assessed on benchmarks ranging from completing a code snippet (White et al., 2025a; Chen et al., 2021) to repairing an issue in a large codebase (Jimenez et al., 2024), and more pertinently, for program synthesis from natural language descriptions (Hendrycks et al., 2021; Jain et al., 2024). Existing benchmarks, however promising, often fall short of capturing the complex nuances of code generation, particularly in assessing how well models align generated code with the developer's instructions.

Among all functionally correct solutions to a problem described in natural language, developers often prefer one particular implementation over others. For instance, when iterating over a list in Python, both a for loop and a list comprehension are functionally correct. Yet, a developer who finds one-liners harder to interpret, may instruct the model to use the for loop for its enhanced readability. Such developer-defined instructions can reflect a variety of concerns, including code refactoring instructions (Fowler, 2018), adherence to stylistic conventions such as Python best practices (van Rossum et al., 2001), or any other non-functional qualities like reliability and maintainability.

To evaluate models' ability to generate code aligned with developer intent, we introduce `CodeAlignBench` —a benchmark specifically designed to assess instruction-following (IF) capabilities in the context of code constraints. `CodeAlignBench` sources its instructions from a

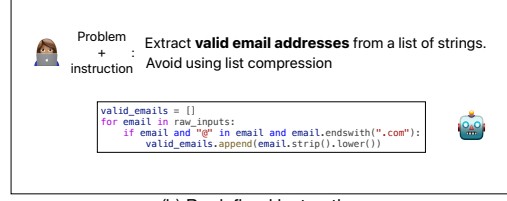

Figure 1: Illustration of two instruction settings in `CodeAlignBench` : (a) Follow-up Instructions, where additional instructions are provided after an initial code generation.(b) Predefined Instructions, where developer constraint is embedded in the initial prompt.

user study with developers across three programming languages, collecting instructions grounded in actual developers preferences. These instruction categories form the foundation for an automated framework that curates IF tasks tailored to each natural language prompt. This framework enables a combination of rule-based methods and LLM-as-a-judge techniques to determine which instruction types apply to a given problem, and to check if the instructions were correctly followed. We empirically evaluated this framework on LLM-judged instructions and found that it achieves a high agreement rate with human judges in verifying whether the instruction was followed, averaging 87%. Overall, this framework enables systematic evaluation of how well models apply developer instructions during code generation.

Figure 1 illustrates the two instruction settings that are supported in this benchmark. In the *Follow-up Instruction* setting, developers provide instructions after the initial code has been generated. In contrast, the *Predefined Instruction* setting embeds the instruction directly within the initial prompt. With model scores differing by about 30% among frontier models, our benchmark provides a meaningful measure of instruction-aligned code generation capabilities, yielding a ranking of models that does not mirror their functional correctness performance. This benchmark also provides a foundation for curating more complex instructions with a combination of atomic instruction in `CodeAlignBench` or iterative refinement in multi-turn settings, allowing models to progressively improve their alignment with developer instructions.

This paper makes the following contributions:

- The first IF benchmark designed to evaluate models ability to generate code aligned with real-world developer instructions.
- An extensive user study with developers across three programming languages to collect and categorize real-world code instructions.
- An automated framework that systematically curates and evaluates IF tasks.
- An empirical evaluation of ten LLMs on `CodeAlignBench` tasks, providing insight into current capabilities and limitations in instruction-aligned code generation.

The remainder of this paper is organized as follows. Section 2 describes the user study and the process of cataloging real-world developer instructions. Section 3 introduces our automated evaluation framework. Section 4 outlines the experimental setup and presents the results. Section 5 discusses related work, and Section 6 concludes the paper with directions for future research.

## 2 INSTRUCTION CATALOG CONSTRUCTION

To construct the instruction catalog, we conducted a user study involving developers with expertise in three different programming languages: Python, Java, JavaScript. For each programming language, participants were presented with pairs of functionally correct code solutions for programming problems from competitive websites and were asked to identify which version they preferred. They were then instructed to write natural language instructions that, if followed, would transform the less preferred code into the more desirable one. These instructions reflect actionable, human-authored guidance aimed at improving code quality, style, readability, or structure — beyond mere functional correctness. Appendix B illustrates an example of such a task. The collected instructions

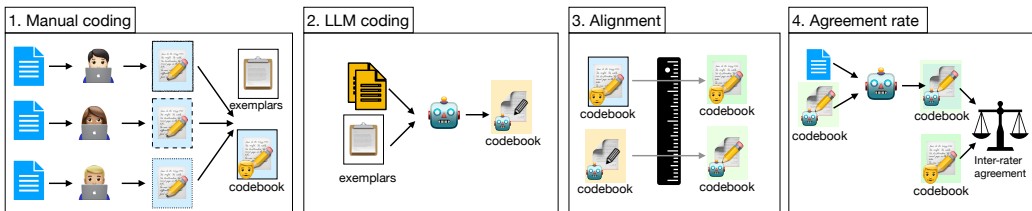

Figure 2: LLM-Assisted Coding Procedure. Stage 1: Manual open coding to create an initial codebook. Stage 2: Exemplar-based prompting of an LLM to generate codes at scale. Stage 3: Alignment and consolidation of LLM- and human-created codebooks. Stage 4: Evaluation of LLM coding reliability against human labels using inter-rater agreement.

were then analyzed through an open coding process, with the assistance of LLMs, to identify common themes and instruction types. This iterative coding process resulted in a structured taxonomy of instructions, which forms the basis of our instruction catalog.

Section 2.1 provides details on the design and implementation of the user study, including task selection and developer guidelines. Section 2.2 outlines the open coding methodology used to synthesize the instruction types, including the role of LLMs in accelerating analysis and ensuring consistency across languages.

## 2.1 USER STUDY DESIGN

**Tasks:** The input set for our user study is derived from the LiveBench (White et al., 2025b) code generation tasks, i.e. programming questions from LeetCode and atCoder. To create pairs of functionally correct code variants for each task, we utilized seven different LLMs to generate code completions. Our goal was to randomly select two generations per task that passed all predefined test cases, ensuring both versions were functionally correct. However, naive random sampling tends to favor higher-performing models, as they produce correct outputs more frequently. To mitigate this sampling bias and ensure fair representation across models, we employed a balanced sampling strategy, ensuring a more balanced distribution of code across the questions.

**Participants and guidelines:** The tasks were conducted by a team of 30 developers, i.e., 10 raters across three programming languages (Python, Java and Javascript). Raters were software developers with experience spanning from 3 to 16 years. Each developer completed all tasks and was asked to choose the code response they preferred, without any additional guidance. This design allowed raters to apply their own criteria for evaluation, such as the readability of code, coding style, or other quality aspects they deemed important. If a code was selected, a text box prompted them to provide one or more instructions they would give to improve the less preferred code. To avoid forcing arbitrary decisions, raters were also given the option to select "no preference", or indicate that they had no clear reason for their preference. This helped ensure that responses were authentic and not influenced by perceived expectations. Additionally, to ensure response quality, five tasks were randomly selected and repeated within the task set. Developers who did not demonstrate consistent preferences across repeated tasks were disqualified, and their responses were excluded from the final analysis.

## 2.2 LLM-ASSISTED CODING

In this section, we describe the process of constructing a catalog of instruction types from raw developer-provided responses. Our methodology is inspired by prior work on human-LLM collaborative coding (Dai et al., 2023), and it follows a four-stage inductive coding process as outlined in Figure 2.

In the first stage, we employed stratified random sampling to select a representative subset of developer responses from each programming language. Using standard statistical sampling techniques, we chose 50 responses per language to achieve a 90% confidence level with a 5% margin of error. These 250 responses were selected for manual open coding. Three authors independently familiarized themselves with the sampled data, coded the instructions using an open coding approach,

Table 1: Summary of instructions by language and type

| Java | JavaScript | Python |
|---|---|---|
| 184 | 175 | 171 |
| Cosmetic | Semantic | Structural |
| 32 | 89 | 104 |
| Performance | Correctness | Algorithm |
| 54 | 6 | 29 |
| Total verified instructions: 228 | | |

and developed an initial codebook of instruction categories. The coders then met to compare and reconcile their individual types of instructions, resolve any disagreements through discussion, and consolidate the instruction categories. From this curated subset, 8 representative examples were selected to serve as exemplars in the next stage. According to previous studies, 4-8 exemplars is optimal for in-context learning (Min et al., 2022). The exemplars described the code for a sample user response and the rationale behind that decision.

In the second stage, we used the selected exemplars to prompt a LLM to code the remaining developer responses. This prompt included the exemplars and a description of the coding task. The LLM returned a proposed codebook and corresponding labels for each instruction in the dataset, effectively simulating a human-led thematic analysis at scale.

In the third stage, we aligned the LLM-generated codebook with the human-created codebook. First, we prompted an LLM to find semantically similar codes. If two codes were semantically similar but differed in phrasing (e.g., "add comments" vs. "enhance documentation"), we manually consolidated them under a single, consistent label. This harmonization step ensured consistent interpretation across sources.

Finally, to assess the quality and reliability of the LLM's categorizations, we prompted it to label the 250 held-out responses from the initial manual coding phase, using the revised codebook. We compared these LLM-assigned codes with human labels and computed inter-rater agreement using Cohen's kappa. This provided a quantitative measure of the LLM's effectiveness and reliability in the coding task. For the Python category, Claude Sonnet 4 achieved a higher agreement rate of 0.75 versus GPT 5 Nano that reached 0.72 and has been used as our assistance. However, the performance of GPT 5 Nano demonstrates that even smaller models can be reliably used in this process.

Above we described our coding process for generating a catalog of diverse instruction categories, and their corresponding real user responses. We also leveraged an LLM to flag user responses that were sufficiently generic to be applicable across multiple problems within the same category. For instance, under the category "use descriptive names," a response like "rename get_y to get_height to be informative" is specific to a particular problem, whereas "use more descriptive names" is generic enough to be applied to other code snippets with the same issue. This step was critical for assigning instructions at the right level of granularity and applicability to our problem statements during benchmarking that we will discuss in the next section. Appendix A provides examples of developer instructions with instruction ids. A full list of instructions are also available in our supplementary materials.

To ensure the integrity of the data, a human verification step was conducted to assess each instruction and filter out developer strings that were overly specific, contradictory, or ambiguous. Human annotators also categorized the instruction categories into the following types:

**Cosmetic:** Modifications affecting readability, style, or presentation without altering the underlying logic.

**Structural:** Changes to the implementation's form or structure (e.g., using specific constructs or data structures) while preserving the core logic.

**Semantic:**

– Algorithm: Tasks requiring a fundamental change in the high-level problem-solving strategy, such as replacing a brute-force approach with a dynamic programming solution.
– Performance: Instructions targeting improvements in time or space complexity by optimizing computations, reducing redundancy, or managing resources more effectively.

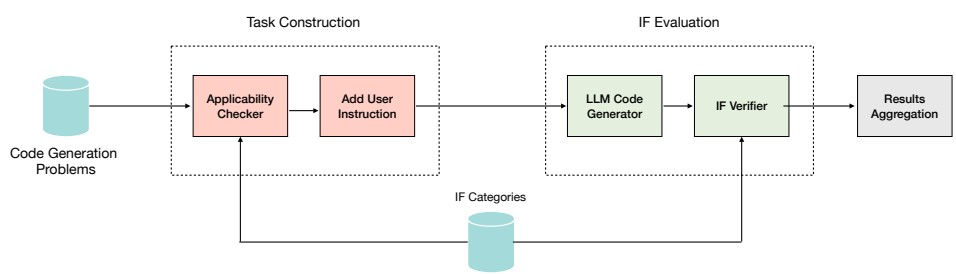

Figure 3: Instruction-following benchmarking framework for code generation

    – Correctness: Instructions aimed at fixing bugs, handling edge cases, and ensuring the code is resilient to unexpected inputs or states.

The detailed guideline on different types is available in our supplementary materials. As shown in Table 1, in total, we cataloged 228 verified instructions, consisting of 32 cosmetic, 89 semantic, and 114 structural ones. This categorization facilitates insights into the instruction types where models perform well or tend to fail. It also enables a uniform distribution of IF tasks when sampled multiple times for benchmarking that will be discussed in details next.

## 3 INSTRUCTION FOLLOWING BENCHMARKING

In this section, we present an automated framework for evaluating the ability of LLMs to follow developer-provided instructions. Figure 3 provides an overview of our benchmarking pipeline, which is organized into two primary stages: Task Construction and IF Evaluation, denoted by the dashed boxes in the figure.

In the Task Construction stage, we curate instruction-following tasks from existing code generation problems and categorize them based on the instruction types extracted through our user study (Section 2). In the subsequent IF Evaluation stage, we evaluate model performance in adhering to these instructions. To support both stages, our framework offers an interface for defining instructions from the catalog. Each instruction includes two key functions: `is_applicable(code)` and `verify(code_after, code_before: Optional)`. The former determines whether the instruction is relevant to a given code snippet, while the latter verifies whether the instruction has been successfully followed. The specific role of these functions and the details of each stage in the pipeline are elaborated in the remainder of this section.

### 3.1 TASK CONSTRUCTION

Our benchmarking pipeline begins by collecting code responses from all models under test, for every problem in the given dataset.

the Task Construction stage creates IF tasks by associating each code solution to a subset of relevant instruction categories along with the user responses for those categories collected in Section 2.

Not all instruction categories are applicable to every code sample. For instance, the category "avoid recursion" is only relevant to recursive implementations and does not apply to iterative ones. To address this, the Applicability Checker identifies a suitable subset of instruction categories for each code sample.

The Applicability Checker identifies $k$ applicable categories by randomly sampling from the catalog. To ensure diversity of instructions within each sample and maintain consistent instruction category distributions across runs, we employ balanced sampling, i.e., drawing from each instruction type described in Section 2. For each type and candidate category, the checker invokes the `is_applicable` function, which is implemented individually for each instruction. This interface encapsulates the logic specific to each instruction, regardless of whether they have been implemented through a rule-based approach or LLM judge, enabling high-accuracy applicability checks that account for both the syntax and semantics of the code. Complete implementations of these functions

for all instruction categories are included in the supplementary materials and will be made available online upon approval.

For each applicable category, a generic developer-provided instruction is sampled and added to the collection of applicable instructions. If no such generic instruction exists for a category, the category id and description are used as a proxy to represent the generic intent of the associated specific instructions. (More details in Section 2).

The collection of applicable instructions are added to the original programming problem to generate IF problems in the IF Evaluation stage as described below.

## 3.2 Instruction Following Evaluation

For each IF problem, we evaluate models under two different settings: (1) providing the coding problem, the initial generated code, and instruction, i.e., Follow-up Instructions (Figure 1 (a)); and (2) providing the coding problem with only the instruction, i.e., Predefined Instructions (Figure 1 (b)). In each setting, we first collect new model responses for the constructed IF problem. Then, an IF Verifier component evaluates whether the provided instruction was successfully followed. This verifier leverages the `verify` function defined for each instruction category, which determines whether the new solution adheres to the instruction. In the first scenario (Follow-up), both the original and the updated solutions are provided for verification, while only the updated solution is available to the second scenario (Predefined). Similar to `is_applicable` function, `verify` can be implemented with any approach, rule-based or LLM judge, tailored for each instruction. Finally, the verifier's binary judgments are aggregated across all tasks to report each model's success rate.

## 4 Experiments

In this section we discuss the details of experiments and present the results of different models on `CodeAlignBench` .

## 4.1 Dataset source

`CodeAlignBench` requires coding problems as a basis for curating IF tasks. In this work, we build on the LiveBench code generation tasks (White et al., 2025b), which consist of coding problems from platforms such as LeetCode and AtCoder. These problems are regularly updated to mitigate risks of data contamination. While the original benchmark only supports Python, we extend it to include additional languages such as Java and JavaScript by implementing an automated translation pipeline and dedicated evaluation environments for each language. This pipeline enables the automatic extension of the latest versions of LiveBench to new languages. The translated dataset, along with details of the translation process and execution environments, is provided in the supplementary materials.

## 4.2 Experimental setup

All experiments were conducted on a local machine, a MacBook Pro equipped with an Apple M1 chip and 16 GB of unified memory. LLM inference for obtaining solutions and judgments were completed using calls to an API. We used this setup to run the evaluation pipeline, including prompt generation, code execution, and automated grading across multiple programming languages. Our modular pipeline is lightweight and platform-agnostic, enabling reproducibility even on resource-constrained machines.

For LLM-assisted annotation, we use OpenAI's GPT o4 mini for creating the aligned codebook and Claude's sonnet 4 to filter user strings. For instructions that use an LLM judge in their applicability and verification implementation, we use Claude Sonnet 4. For benchmarking, we evaluate 10 models spanning 3 major families, OpenAI, Claude and Gemini, to represent a diverse set of design trade-offs. These models vary in architecture, parameter scale, reasoning depth, and release recency, ensuring coverage of both state-of-the-art leading models and smaller, more cost-efficient alternatives.

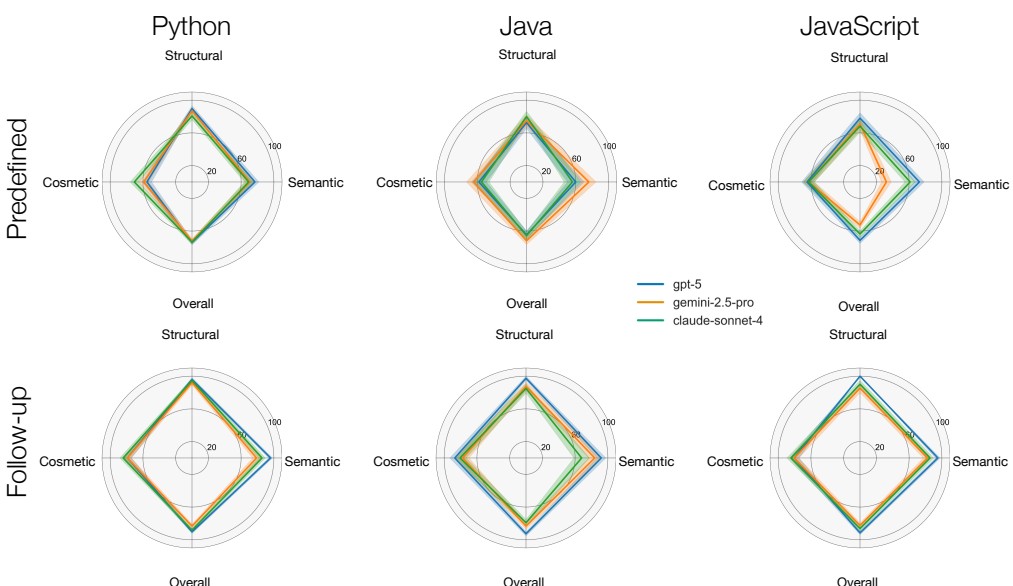

Figure 4: Radar plots of the top models from each family (GPT, Gemini, and Sonnet), showing performance across instruction categories (Structural, Semantic, and Cosmetic) as well as the overall aggregate. The top panels correspond to predefined tasks, while the bottom panel presents follow-up tasks. Shaded regions represent the standard error of the mean (SEM).

### 4.3 BENCHMARK RESULTS

A primary finding of our study is the significant performance gap between task types, where models consistently performed better on follow-up tasks than on predefined tasks. This observation held true across all evaluated languages and was confirmed by Wilcoxon signed-rank tests, which revealed a substantial improvement for follow-up tasks in Python (median difference = 0.181, $p < 0.01$), Java (median difference = 0.146, $p < 0.01$), and JavaScript (median difference = 0.240, $p < 0.01$). This confirms that providing contextual history allows models to better follow user instructions, and generate more accurate code modifications. Alternatively, this suggests that models are more effective with a single, focused instruction than with a problem–instruction pair, which may constitute a more complex task.

Beyond the influence of scenarios, the nature of the required modification also proved to be a critical factor. Structural tasks, in particular, consistently yielded the highest performance scores. A Friedman test substantiated this finding for predefined tasks across Python ($\chi^2(2) = 12.20, p = 0.0022$, Kendall's $W = 0.61$), Java ($\chi^2(2) = 6.20, p = 0.045, W = 0.31$), and JavaScript ($\chi^2(2) = 12.60, p = 0.0018, W = 0.63$). A similar trend was observed for follow-up tasks, where structural modifications led to significantly higher performance in JavaScript ($\chi^2(2) = 9.80, p = 0.0074$, $W = 0.49$).

In terms of model-specific performance, GPT-5, GPT-5 mini, two latest members of the openAI family, and GPT-o3 mini with high reasoning emerged as the top-performing models, though their results were often statistically indistinguishable from one another (Table 2). Notably, these leading GPT models significantly outperformed the best models from the Sonnet and Gemini families in several instances. This pattern of dominance was less pronounced, however, when analyzing the granular Semantic, Structural, and Cosmetic categories, where significant differences between top-tier models were rare.

A secondary trend observed was consistent generational improvement within model families. For example, Sonnet 4 surpassed Sonnet 3.7 across all evaluated categories and languages, illustrating a clear pattern of iterative refinement. However, these individual improvements were not always statistically significant across all model families.

Finally, despite the notably strong performance of GPT models, our results indicate that no model achieved performance saturation across the full suite of tasks. Even the most capable models exhib-

ited clear limitations particularly in the semantic and cosmetic categories. This substantial headroom for improvement, along with the flexibility of our benchmarking pipeline where we are able to combine multiple instructions and tune task difficulties, underscores the utility of our benchmark for tracking future progress.

Table 2: Model performance for different instruction categories across languages. Left-hand values in each column are for predefined scenarios; right-hand values are for follow-up scenarios. Greyed values indicate the confidence-interval margins (i.e., the rates are within ± the grey value). Gem = Gemini, Fl = Flash, Pr = Pro, mi = mini

| Model | Semantic | Structural | Cosmetic | Overall |
|---|---|---|---|---|
| **Python** | | | | |
| Sonnet 3.7 | 0.40 (0.11) / 0.53 (0.11) | 0.49 (0.11) / 0.61 (0.11) | 0.54 (0.11) / 0.79 (0.09) | 0.47 (0.06) / 0.64 (0.06) |
| Sonnet 4 | 0.69 (0.10) / 0.86 (0.08) | 0.81 (0.09) / 0.95 (0.05) | 0.71 (0.10) / 0.84 (0.08) | 0.74 (0.06) / 0.88 (0.04) |
| Gem 2.0 Fl | 0.31 (0.10) / 0.43 (0.11) | 0.51 (0.11) / 0.63 (0.11) | 0.42 (0.11) / 0.80 (0.09) | 0.41 (0.06) / 0.62 (0.06) |
| Gem 2.5 Fl | 0.45 (0.11) / 0.63 (0.11) | 0.73 (0.10) / 0.87 (0.08) | 0.47 (0.11) / 0.72 (0.10) | 0.55 (0.06) / 0.74 (0.06) |
| Gem 2.5 Pr | 0.71 (0.10) / 0.79 (0.09) | 0.87 (0.07) / 0.92 (0.06) | 0.59 (0.11) / 0.79 (0.09) | 0.72 (0.06) / 0.83 (0.05) |
| GPT 4.1 | 0.60 (0.11) / 0.84 (0.08) | 0.74 (0.10) / 0.87 (0.08) | 0.50 (0.11) / 0.87 (0.08) | 0.62 (0.06) / 0.86 (0.05) |
| GPT 4o | 0.37 (0.11) / 0.80 (0.09) | 0.65 (0.11) / 0.82 (0.09) | 0.51 (0.11) / 0.87 (0.07) | 0.51 (0.06) / 0.83 (0.05) |
| GPT 5 | 0.77 (0.09) / 0.96 (0.04) | 0.90 (0.07) / 0.96 (0.04) | 0.55 (0.11) / 0.79 (0.09) | 0.74 (0.06) / 0.90 (0.04) |
| GPT 5 mini | 0.73 (0.10) / 0.93 (0.06) | 0.83 (0.08) / 0.97 (0.04) | 0.63 (0.11) / 0.80 (0.09) | 0.73 (0.06) / 0.90 (0.04) |
| GPT o3 mi | 0.65 (0.11) / 0.95 (0.05) | 0.83 (0.08) / 0.95 (0.05) | 0.60 (0.11) / 0.88 (0.07) | 0.70 (0.06) / 0.93 (0.03) |
| **Java** | | | | |
| Sonnet 3.7 | 0.48 (0.20) / 0.40 (0.20) | 0.57 (0.16) / 0.60 (0.15) | 0.55 (0.18) / 0.71 (0.16) | 0.54 (0.10) / 0.58 (0.10) |
| Sonnet 4 | 0.56 (0.20) / 0.68 (0.19) | 0.80 (0.13) / 0.85 (0.11) | 0.55 (0.18) / 0.81 (0.14) | 0.66 (0.10) / 0.79 (0.08) |
| Gem 2.0 Fl | 0.44 (0.20) / 0.52 (0.20) | 0.45 (0.16) / 0.45 (0.16) | 0.61 (0.17) / 0.68 (0.17) | 0.50 (0.10) / 0.54 (0.10) |
| Gem 2.5 Fl | 0.60 (0.20) / 0.68 (0.19) | 0.50 (0.16) / 0.68 (0.15) | 0.48 (0.18) / 0.68 (0.17) | 0.52 (0.10) / 0.68 (0.09) |
| Gem 2.5 Pr | 0.76 (0.17) / 0.84 (0.15) | 0.75 (0.14) / 0.88 (0.10) | 0.65 (0.17) / 0.77 (0.15) | 0.72 (0.09) / 0.83 (0.07) |
| GPT 4.1 | 0.64 (0.19) / 0.72 (0.18) | 0.75 (0.14) / 0.88 (0.10) | 0.52 (0.18) / 0.87 (0.12) | 0.65 (0.10) / 0.83 (0.07) |
| GPT 4o | 0.48 (0.20) / 0.56 (0.20) | 0.55 (0.16) / 0.78 (0.13) | 0.58 (0.18) / 0.84 (0.13) | 0.54 (0.10) / 0.74 (0.09) |
| GPT 5 | 0.60 (0.20) / 0.92 (0.11) | 0.72 (0.14) / 0.97 (0.05) | 0.58 (0.18) / 0.87 (0.12) | 0.65 (0.10) / 0.93 (0.05) |
| GPT 5 mi | 0.72 (0.18) / 0.80 (0.16) | 0.88 (0.10) / 0.93 (0.08) | 0.65 (0.17) / 0.90 (0.11) | 0.76 (0.09) / 0.89 (0.06) |
| GPT o3 mi | 0.64 (0.19) / 0.76 (0.17) | 0.65 (0.15) / 0.93 (0.08) | 0.58 (0.18) / 0.94 (0.09) | 0.62 (0.10) / 0.89 (0.06) |
| **JavaScript** | | | | |
| Sonnet 3.7 | 0.29 (0.11) / 0.56 (0.12) | 0.59 (0.15) / 0.63 (0.15) | 0.41 (0.14) / 0.69 (0.13) | 0.41 (0.08) / 0.62 (0.08) |
| Sonnet 4 | 0.61 (0.12) / 0.85 (0.09) | 0.68 (0.14) / 0.90 (0.09) | 0.63 (0.13) / 0.84 (0.10) | 0.64 (0.08) / 0.86 (0.05) |
| Gem 2.0 Fl | 0.27 (0.11) / 0.47 (0.13) | 0.46 (0.15) / 0.71 (0.14) | 0.47 (0.14) / 0.78 (0.11) | 0.39 (0.08) / 0.64 (0.08) |
| Gem 2.5 Fl | 0.37 (0.12) / 0.66 (0.12) | 0.56 (0.15) / 0.93 (0.08) | 0.49 (0.14) / 0.78 (0.11) | 0.46 (0.08) / 0.77 (0.07) |
| Gem 2.5 Pr | 0.32 (0.12) / 0.82 (0.10) | 0.71 (0.14) / 0.85 (0.11) | 0.63 (0.13) / 0.80 (0.11) | 0.53 (0.08) / 0.82 (0.06) |
| GPT 4.1 | 0.65 (0.12) / 0.85 (0.09) | 0.73 (0.14) / 0.93 (0.08) | 0.45 (0.14) / 0.75 (0.12) | 0.60 (0.08) / 0.84 (0.06) |
| GPT 4o | 0.39 (0.12) / 0.76 (0.11) | 0.73 (0.14) / 0.93 (0.08) | 0.51 (0.14) / 0.73 (0.12) | 0.52 (0.08) / 0.79 (0.06) |
| GPT 5 | 0.73 (0.11) / 0.95 (0.05) | 0.78 (0.13) / 1.00 (0.00) | 0.65 (0.13) / 0.80 (0.11) | 0.71 (0.07) / 0.92 (0.04) |
| GPT 5 mi | 0.71 (0.11) / 0.90 (0.07) | 0.88 (0.10) / 0.95 (0.07) | 0.49 (0.14) / 0.82 (0.11) | 0.68 (0.07) / 0.89 (0.05) |
| GPT o3 mi | 0.66 (0.12) / 0.90 (0.07) | 0.80 (0.12) / 1.00 (0.00) | 0.49 (0.14) / 0.86 (0.10) | 0.64 (0.08) / 0.92 (0.04) |

**LLM Judgment reliability:** LLM judgments were used to determine whether a given instruction was correctly followed. To evaluate our model's performance on this task, we employed a set of questions containing real user instructions collected from the user study. This approach allowed us to isolate the verifier component from the rest of the pipeline (i.e., the applicability checker) and focus solely on its effectiveness. We picked top 10 most frequent instructions, least 10 frequent ones, as well as 10 randomly selected ones. This ensured that the evaluation covered both commonly encountered instructions and those that might be unique to a particular user or code snippet. Then, two authors independently assessed the responses and resolved any disagreements through discussion to reach a consensus. We compared the outputs of Claude Sonnet 4 performing the same verification task against human assessments. Using Claude Sonnet 4, we observed four instances of disagreement with human judgments, giving this model 86.67 accuracy in terms of verifying if the instruction has been followed. We also ran the same set of questions from the user study with Gemini 2.5 pro. We found that 0.8249 of Gemini 2.5 pro's judgments matched the predictions from Claude Sonnet 4. Further analysis of the LLM judge can be found in Appendix D. These results demonstrate the reliability of LLMs, under our prompting, in performing instruction following verification.

## 5 RELATED WORKS

**Evaluating Code Generation Models:** Code generation using LLMs has become a prominent area of research and practical application (Jiang et al., 2024; Jain et al., 2024). By leveraging representations from vast pre-training corpora, LLMs have demonstrated a remarkable capability to transform natural language descriptions into functional code (Hou et al., 2024). This progress has driven the creation of benchmarks to systematically evaluate reliability and guide future work (Jiang et al., 2024). The majority of these benchmarks consist of short, self-contained, algorithmic tasks, exemplified by HumanEval (Chen et al., 2021) and MBPP (Austin et al., 2021). Recent benchmarks have expanded the scope of evaluation to more complex settings, such as class-level generation (Du et al., 2023) and repository-level tasks built on open-source projects (Li et al., 2024; Liu et al., 2024). Unlike `CodeAlignBench`, these benchmarks often stop at the threshold of functional correctness, content with any working solution in the vast space of possible solutions.

**Instruction Following Evaluation:** IF is a fundamental capability of LLMs. To evaluate this ability systematically, researchers have proposed a growing number of automated benchmarks. Most of these approaches define instruction following in terms of format adherence; a model is considered successful when its outputs meet the explicit requirements specified in the prompt.

Early efforts in evaluating IF, such as IFEval (Zhou et al., 2023; He et al., 2024), focused on automatically verifiable constraints like word limits or required phrases, which allowed for straightforward, rule-based evaluation. Building on this foundation, more recent work like LiveBench has extended this paradigm to diverse creative tasks such as summarization and story generation, while still maintaining the format of objective checks against explicit instructions (White et al., 2025b). Alongside increasing task diversity, research has also broadened the scope of IF benchmarks. For instance, multilingual variants like M-IFEval (Dussolle et al., 2025) highlight performance variations across languages, while LIFBench addresses the challenges of instruction following in long-context settings (Wu et al., 2025). A complementary line of inquiry examines the evaluation process itself, with approaches like LLMBar (Zeng et al., 2024) assessing the ability of LLMs to act as reliable judges. Collectively, these diverse efforts are crucial for ensuring that IF benchmarks move beyond simple instructions in unrealistic settings to become truly informative and robust.

Although most IF benchmarks have centered on natural language tasks, more recent work extends these evaluation to the code generation setting. Early efforts in this space, such as CodeIF (Wang et al., 2025a) introduced multi-constraint tasks across multiple languages and assesses adherence with automated rule-based metrics. BigCodeBench-Instruct (Zhuo et al., 2025) addresses instruction following in more complex contexts by reformulating function docstrings into concise and verifiable natural language tasks that may require multiple library calls. CodeIF-Bench (Wang et al., 2025b) aims to simulate developer workflows in interactive settings by means of verifiable instructions and test-case scoring; however, its tasks are synthetically constructed rather than sourced from real developers . Complementing these constraint-centric efforts, CodeUltraFeedback (Weyssow et al., 2024) shifts the focus from correctness to quality, leveraging an LLM to judge preferences like readability and style. Despite these advances, a common limitation persists: many of these benchmarks rely on synthetic tasks and prioritize easily verifiable outcomes. Our work builds on this foundation by grounding evaluation in realistic instructions sourced directly from developers. Moreover, `CodeAlignBench` reduces the risk of data contamination, an issue for static benchmarks that often appear in the training data of newer models, by incorporating live benchmarks such as LiveBench (White et al., 2025b), which are continuously updated with new, real-world tasks.

## 6 CONCLUSION

In this paper we introduced a comprehensive and extensible benchmark for LLM code generation. It extends an existing dataset to support three languages and a catalog of instruction-based tasks, evaluating dimensions beyond functional correctness. Our findings show that frontier models, while often generating functionally correct code, fail to adhere to specific stylistic or structural instructions. The novel, modular design of our evaluation pipeline is central to our contribution. This paves the way for more nuanced evaluations by enabling the seamless addition of new problems and, in the future, more complex multi-turn instructions. To ensure clarity and correctness, we also employed LLMs to proofread the written text and refine grammar throughout the paper.

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

# A SAMPLE OF CODEALIGNBENCH 'S INSTRUCTIONS

Table 3: Sample of developer instructions with language column.

| Category | Instruction ID | Sample User String | Language |
|---|---|---|---|
| **Cosmetic** | add-comments | Add comments to complex main logic of the code. | JS, P, J |
| | add-type-hints | Add type annotations to function parameters and return values. | P |
| | concise-variable-names | Use concise variable naming. | JS, P, J |
| | consistent-formatting | Use consistent formatting for spacing and variables. | JS, P, J |
| | consistent-naming-convention | Use consistent naming convention. | JS, P, J |
| | move-imports-to-top | Move imports to the top of the file. | JS, P, J |
| | reduce-comments | Remove unnecessary comments from the code. | JS, P, J |
| | remove-redundant-code-and-comments | Remove redundant comments and variables. | JS, P, J |
| | remove-unused-imports | Remove unused import statements. | JS, P, J |
| | use-lowercase-variable-names | Use lowercase variable names. | JS, P, J |
| | use-symbolic-constants | Use symbolic constants. | JS, P, J |
| **Structural** | add-intermediate-variable | Store intermediate results in a separate variable. | JS, P, J |
| | add-missing-import | Identify and add missing import statements. | JS, P, J |
| | add-test-cases | Create executable test code to verify correctness. | JS, P, J |
| | avoid-code-duplication | Eliminate repeated code by consolidating duplicate logic. | JS, P, J |
| | avoid-global-variables | Avoid using global or external variables. | JS, P, J |
| | avoid-nested-functions | Avoid defining functions inside other functions. | JS, P |
| | inline-helper-functions | Inline simple helper functions. | JS, P, J |
| | modularize-code | Use proper functions or modules for the logic. | JS, P, J |
| | move-print-outside-loop | Produce output at the end of the main code logic. | JS, P, J |
| | remove-unused-function | Remove unused functions. | JS, P, J |
| | shorten-if-else-chains | Reduce the length of if-else chains. | JS, P, J |
| | use-counter-object | Use Counter for frequency counting. | P |
| | use-dictionary-mapping | Use a dictionary for mapping. | JS, P, J |
| | use-enumerate | Use enumerate for loops. | JS, P |
| | use-list-comprehensions | Use list comprehensions. | P |
| | use-loop-indices | Use index-based traversal. | JS, P, J |
| | use-string-multiplication-operator | Use string multiplication operator. | P |
| **Semantic** | | | |
| Algorithm | avoid-floating-point-operations | Avoid floating-point operations. | JS, P, J |
| | avoid-recursion | Avoid using recursion. | JS, P, J |
| | change-inner-loop-to-start-from-outer-index | Change inner loop to start from outer index. | JS, P, J |
| | use-binary-search | Implement binary search for value lookups. | JS, P, J |
| | use-dynamic-programming | Use dynamic programming. | JS, P, J |
| | use-greedy-algorithm | Use greedy algorithm. | JS, P, J |
| | use-iterative-dp | Use iterative dynamic programming. | JS, P, J |
| Performance | avoid-extra-array-storage | Avoid unnecessary array storage. | JS, P, J |
| | avoid-redundant-computation | Avoid redundant computation. | JS, P, J |
| | limit-variable-scope | Limit the tracking scope or prune the search space. | JS, P, J |
| | optimize-io-performance | Use buffered I/O and input processing. | JS, P, J |
| | reduce-dictionary-operations | Reduce dictionary operations. | JS, P, J |
| | set-recursion-limit | Set recursion limit. | P |
| | use-efficient-algorithm | Use a more efficient algorithm. | JS, P, J |
| | use-efficient-data-structure | Use a more efficient data structure. | JS, P, J |
| Correctness | check-edge-cases | Add checks for edge cases when possible. | JS, P, J |
| | error-handling | Handle different errors properly. | JS, P, J |

# B  SAMPLE OF USER STUDY TASKS

Imagine you are working with a code-generating chatbot like ChatGPT. You input a natural language request describing a coding task and you receive two code snippets, both from different models. Your task will be to choose which generation you prefer.

| Left | Right |
|---|---|

```python
class Solution:
    def numberGame(self, nums: List[int]) -> List[int]:
        arr = []
        nums_sorted = sorted(nums)
        for i in range(0, len(nums_sorted), 2):
            # Bob appends first
            arr.append(nums_sorted[i+1])
            # Alice appends next
            arr.append(nums_sorted[i])
        return arr
```

```python
class Solution:
    def numberGame(self, nums: List[int]) -> List[int]:
        nums.sort()
        arr = []
        for i in range(0, len(nums), 2):
            arr.append(nums[i+1])
            arr.append(nums[i])
        return arr
```

Think about the kind of code that you like to see. Decide on which model generation you prefer.

Which model generation do you prefer?
○ Left
○ Right
○ No Preference

For the model you do *not* prefer, think about instructions you can provide that will make your least preferred model more like your most preferred.         ☐ Unsure about preference reason
Provide one or more instructions to improve the least preferred model. If you have more than one instruction, use the "add" button to add the additional instructions.

Figure 5: Pairwise evaluation task

## C  SAMPLE SCREENSHOT OF PAIRWISE MODEL COMPARISON

The benchmark is accompanied by visualization tools designed to aid in the interpretation and analysis of results. Image 6 illustrates a screenshot of pairwise comparator.

**Pairwise Analysis**

**Previous solution**

```python
class Solution:
    def findWordsContaining(self, words: List[str], x: str) -> List[int]:
        result = []

        for i, word in enumerate(words):
            if x in word:
                result.append(i)

        return result
```

**ID:** 9026b23d1094bb0072bae31bd66f9a8cc7220e3951dc520c53a2f21c4e8a472c

**Category:** add-test-cases

**Result:** claude-sonnet-4 fails, gpt-5-mini passes

Instruction: Add test cases as executable examples to validate core logic

**Claude Sonnet 4**

```python
class Solution:
    def findWordsContaining(self, words: List[str], x: str) -> List[int]:
        result = []

        for i, word in enumerate(words):
            if x in word:
                result.append(i)

        return result
```

**GPT-5-mini**

```python
class Solution:
    def findWordsContaining(self, words: List[str], x: str) -> List[int]:
        result = []

        for i, word in enumerate(words):
            if x in word:
                result.append(i)

        return result

if __name__ == "__main__":
    sol = Solution()

    # Example 1
    words1 = ["leet", "code"]
    x1 = "e"
    out1 = sol.findWordsContaining(words1, x1)
    print("Output 1:", out1)
    assert out1 == [0, 1], f"Expected [0, 1], got {out1}"

    # Example 2
    words2 = ["abc", "bcd", "aaaa", "cbc"]
    x2 = "a"
    out2 = sol.findWordsContaining(words2, x2)
    print("Output 2:", out2)
    assert out2 == [0, 2], f"Expected [0, 2], got {out2}"

    # Example 3
    words3 = ["abc", "bcd", "aaaa", "cbc"]
    x3 = "z"
    out3 = sol.findWordsContaining(words3, x3)
    print("Output 3:", out3)
    assert out3 == [], f"Expected [], got {out3}"

    # Additional tests
    words4 = ["a", "b", "c", "a"]
    x4 = "a"
    out4 = sol.findWordsContaining(words4, x4)
    print("Output 4:", out4)
    assert out4 == [0, 3], f"Expected [0, 3], got {out4}"

    words5 = ["xyz", "yzx", "zxy"]
    x5 = "x"
    out5 = sol.findWordsContaining(words5, x5)
    print("Output 5:", out5)
    assert out5 == [0, 1, 2], f"Expected [0, 1, 2], got {out5}"

    print("All tests passed.")
```

Figure 6: Example of an instruction that Sonnet 4 failed to follow, but was correctly executed by GPT-5 mini.

# D    RADAR PLOTS FOR ALL MODEL FAMILIES

Here we demonstrate performance of all models within each model family on our benchmark. Substantial and significant generational improvements are observable within different model families.

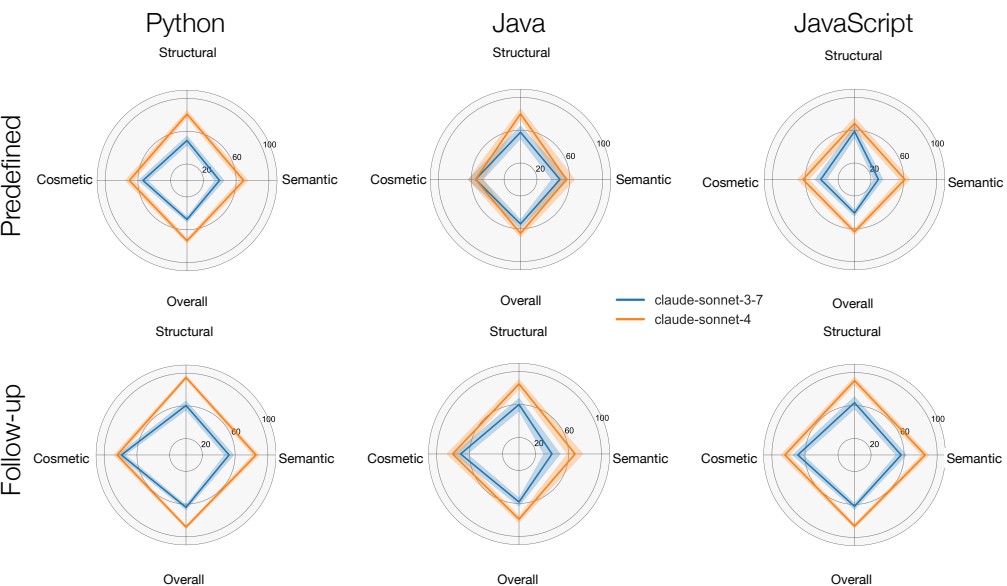

Figure 7: Radar plots of models from the Sonnet family, showing performance across instruction categories (Structural, Semantic, and Cosmetic) as well as the overall aggregate. The top panels correspond to predefined tasks, while the bottom panel presents follow-up tasks. Shaded regions represent the standard error of the mean (SEM).

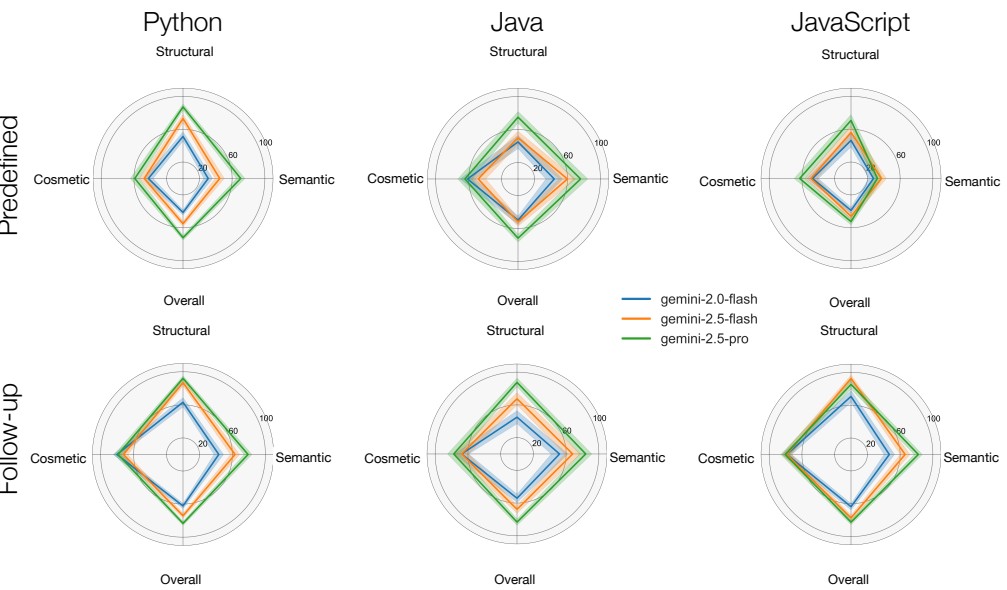

Figure 8: Radar plots of models from the Gemini family, showing performance across instruction categories (Structural, Semantic, and Cosmetic) as well as the overall aggregate. The top panels correspond to predefined tasks, while the bottom panel presents follow-up tasks. Shaded regions represent the standard error of the mean (SEM).

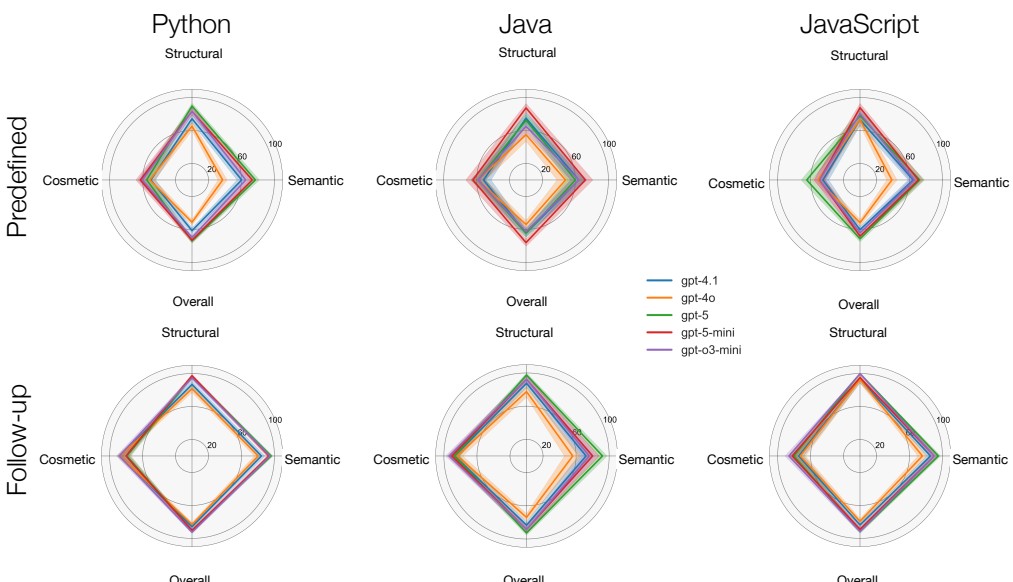

Figure 9: Radar plots of models from the GPT family, showing performance across instruction categories (Structural, Semantic, and Cosmetic) as well as the overall aggregate. The top panels correspond to predefined tasks, while the bottom panel presents follow-up tasks. Shaded regions represent the standard error of the mean (SEM).

# E    LLM JUDGE BIAS

In our experiments, we employ Claude Sonnet 4 as the LLM judge for instruction verification. This creates a potential source of bias as the judgments of Claude Sonnet 4 may be biased towards generations from Claude Sonnet 4. To analyze the effects of using Claude as the judge, we performed an ablation using GPT-5 as the judge model on JavaScript and Java problems. We found that the Sonnet 4 Judge scores are not always higher for Sonnet 4 code generations. Additionally, the relative rankings between models remained pretty consistent.

| Language | Setting | Judge | gemini-2.5-pro | sonnet 4 |
|---|---|---|---|---|
| JavaScript | Follow-up | gpt 5 | 0.790 | 0.811 |
| JavaScript | Follow-up | sonnet 4 | 0.854 | 0.871 |
| JavaScript | Predefined | gpt 5 | 0.602 | 0.615 |
| JavaScript | Predefined | sonnet 4 | 0.585 | 0.671 |
| Java | Follow-up | gpt 5 | 0.794 | 0.811 |
| Java | Follow-up | sonnet 4 | 0.863 | 0.863 |
| Java | Predefined | gpt 5 | 0.653 | 0.670 |
| Java | Predefined | sonnet 4 | 0.649 | 0.632 |

Table 4: Comparison of gemini-2.5-pro and sonnet 4 across different languages, settings, and judges.

