# OpenReview forum: "CODEALIGNBENCH: Assessing Code Generation Models on Developer-Preferred Code Adjustments"
_ICLR.cc/2026/Conference — Submitted to ICLR 2026_

### Official Review · Reviewer_YHLd · 2025-10-27

**Soundness:** 2
**Presentation:** 1
**Contribution:** 2
**Rating:** 2
**Confidence:** 5

**Summary:**

This paper introduces CodeAlignBench, a benchmark designed to evaluate LLMs on their ability to generate code aligned with developer preferences beyond functional correctness. The authors address a critical gap in existing benchmarks by focusing on instruction-following capabilities in two settings: adherence to predefined constraints and refinement via follow-up instructions. The benchmark is built on a user study with developers across Python, Java, and JavaScript, resulting in a catalog of 228 verified instructions categorized into cosmetic, structural, and semantic types. The evaluation pipeline combines rule-based checks and LLM-as-a-judge verification, achieving 87% agreement with human judgments. Empirical results reveal significant performance variations across models, languages, and instruction types, with follow-up tasks consistently outperforming predefined ones.

**Strengths:**

- The paper introduces and evaluates two complementary instruction-following settings, predefined and follow-up, that have not been explored before.
- The authors source instruction categories from realistic grounding, thereby reducing synthetic bias, and construction of an instruction catalog via human–LLM collaborative coding.
- The paper  extends LiveBench to Java and JavaScript, enhancing its cross-lingual relevance and applicability.
- Quantitative results across three languages and 10 models, with appropriate non-parametric tests (Wilcoxon, Friedman) to support main claims.
- Automated translation from Python to Java/JS is helpful, but the fidelity of translated tasks, idiomaticity, and potential language-specific pitfalls are not deeply analyzed. Any translation artifacts could affect applicability checks and success rates.
- The pipeline evaluates whether instructions are “followed,” but it is not always clear how conflicts between functional correctness and instruction adherence are resolved or reported. For example, what if following an instruction degrades correctness or violates hidden constraints?

**Weaknesses:**

- The manuscript contains numerous minor typographical and style inconsistencies that detract from polish and readability. Examples include:
  - Page 1, line 53: the em dash around “CodeAlignBench —a benchmark” has asymmetric spacing (space before the dash, none after).
  - Page 3, line 140: "Javascript" should be "JavaScript" for consistency with other instances.
  - Page 5, line 257: the sentence starts with a lowercase letter; initial capitalization is expected.
- Algorithm questions from LiveBench are not developer-aligned, making the authors' claim unconvincing. Programming tasks from BigCodeBench are more reasonable in this case.
- Limited visibility into false positive/negative patterns for the verifier across instruction types, and how judge errors might differentially impact categories.

**Questions:**

- Does the Fig 1 example come from CodeAlignBench? Is this an illustration only?
- Do you simultaneously check functional correctness after applying instructions? If so, how often do models follow instructions but break correctness, and how is this treated in scoring?
- How was is_applicable accuracy validated? Please report precision/recall against human applicability labels for a sample across languages and instruction types.
- Could you quantify translation fidelity (e.g., manual audits of a sample for idiomaticity and equivalence)? Are there examples where language-specific idioms change the intended instruction (e.g., Python list comprehensions vs. Java streams)?
- Have you considered extending CodeAlignBench to multi-turn developer interactions (beyond single refinement)?
- Given that several evaluated models are trained on large-scale internet corpora that may contain LiveBench, how did you assess and mitigate potential data contamination for CodeAlignBench?

---

### Official Review · Reviewer_oiDR · 2025-11-01

**Soundness:** 2
**Presentation:** 2
**Contribution:** 2
**Rating:** 4
**Confidence:** 3

**Summary:**

This paper proposes CodeAlignBench, which is a new benchmark targeted at evaluating the instruction-following capabilities of LLMs on code generation tasks. Specifically, CodeAlignBench focuses on not only predefined Instructions but also follow-up instructions, providing a more comprehensive evaluation of instruction-following capabilities. The evaluation includes a large number of LLMs from multiple model serieses.

**Strengths:**

- This paper explores an important and interesting research direction.

**Weaknesses:**

- The evaluation lacks detailed analysis and concrete examples about LLMs’ performance on this benchmark. While there is some high-level discussion in Section 4, more in-depth study is required for a better understanding of the instruction-following capabilities of LLMs.
- The evaluation of instruction-following capability is largely dependent on LLM-as-a-Judge, which is however not accurate enough. As shown in Section 4, the accuracy of LLM-as-a-Judge is only 86.67%, which limits its practical usage in the real world.

**Questions:**

- Why is CodeAlignBench constructed based on LiveBench rather than more widely-use coding benchmarks like LiveCodeBench?

---

### Official Review · Reviewer_APFm · 2025-11-04

**Soundness:** 3
**Presentation:** 1
**Contribution:** 2
**Rating:** 4
**Confidence:** 2

**Summary:**

The paper presents a benchmark on instruction following for coding tasks. They evaluate two aspects: following constraints given with the problem and making refinements based on follow-up instructions. Using problems sourced from LiveBench, they study the performance of various models on Python, Java, and JavaScript, showing that models have varying performance on different types of instruction-following beyond just functional correctness, including different adjustment instruction types: semantic, structural, performance, etc.

**Strengths:**

- The process to collect the instruction catalog involves human developers and not just synthetic generation from LLMs, making it a valuable dataset and benchmark
- The motivation of the paper of going beyond functional correctness is well-motivated, as code generation for users is not just about correctness but also about adhering to users' preferences
- Evaluates multiple programming languages (Python, Java, JavaScript), allowing for cross-language comparison

**Weaknesses:**

- Since some instructions can involve non-trivial adjustments to the code on a semantic level, it is not clear how accurate LLM-as-judge is for those kinds of instructions
- Even though the benchmark creation process involves developers, there is still use of LLMs; it would be better if the benchmark could be vetted by humans to ensure validity
- The models used in the experiments are all closed models and no open models are evaluated

**Questions:**

- In 4.3, it is concluded that models in general perform worse on predefined tasks than instruction-following tasks. Do reasoning models reduce this gap? Also, can better prompting help, e.g., asking the model to reflect on the first draft of generated code and revise accordingly?
- Figure 4 radar map is hard to read. Is there a better way to show the insights here? Table 2 is also hard to read - would making the better results bold help?
- How does the judge determine the score? How does scoring work for a response which aligns with the adjustment instruction but makes the code incorrect, compared to a response that doesn't adhere well to the instruction but is still functionally correct?

---

### Meta-Review · Area_Chair_p4z7 · 2026-01-01

**Summary:**

Issue 1: **Potential robustness issue with the LLM judge (APFm, oiDR)** With an accuracy around 86.7, it could potentially “limit its practical usage in the real world” (oiDR). Reviewer YHLd also flagged that there is a lack of analysis on the judge’s performance under different task categories.

Issue 2: **Validity of the benchmark and lack of detailed evaluation** the proposed LLM-in-the-loop benchmark creation process might introduce potential biases (APFm), and no open models are evaluated (APFm). Reviewer oiDR also flagged that there is a lack of detailed evaluation.

**Reviewer Concerns:**

Both issues remain unaddressed as the authors didn't submit a rebuttal.

**Reviewer Scores:**

N/A

---

### Decision · Program_Chairs · 2026-01-26

Reject